# An Interoperable Component-Based Architecture for Data-Driven IoT System

**DOI:** 10.3390/s19204354

**Published:** 2019-10-09

**Authors:** Sin Kit Lo, Chee Sun Liew, Kok Soon Tey, Saad Mekhilef

**Affiliations:** 1Department of Computer System and Technology, Faculty of Computer Science and Information Technology, University of Malaya, Kuala Lumpur 50603, Malaysia; 2Power Electronics and Renewable Energy Research Laboratory (PEARL), Department of Electrical Engineering, Faculty of Engineering, University of Malaya, Kuala Lumpur 50603, Malaysia; saad@um.edu.my

**Keywords:** Internet of Things (IoT), edge computing, system interoperability, component-based design methodology, data-driven mechanism

## Abstract

The advancement of the Internet of Things (IoT) as a solution in diverse application domains has nurtured the expansion in the number of devices and data volume. Multiple platforms and protocols have been introduced and resulted in high device ubiquity and heterogeneity. However, currently available IoT architectures face challenges to accommodate the diversity in IoT devices or services operating under different operating systems and protocols. In this paper, we propose a new IoT architecture that utilizes the component-based design approach to create and define the loosely-coupled, standalone but interoperable service components for IoT systems. Furthermore, a data-driven feedback function is included as a key feature of the proposed architecture to enable a greater degree of system automation and to reduce the dependency on mankind for data analysis and decision-making. The proposed architecture aims to tackle device interoperability, system reusability and the lack of data-driven functionality issues. Using a real-world use case on a proof-of-concept prototype, we examined the viability and usability of the proposed architecture.

## 1. Introduction

The advancement in wireless communication technology and high-speed Internet connectivity have fostered the development of data ubiquity and data heterogeneity. The Internet of Things (IoT) is by far the most idealistic and revolutionary implementation of ubiquitous and heterogeneous computing and, in recent years, the application of the IoT has expanded to various fields, following the introduction of Industry 4.0 [1,2]. According to *Gartner* [3], in 2020, the forecasted connected things will reach 20.4 billion and the total spending on hardware will reach $3 trillion. As the deployment of IoT systems stretches across multiple domains, the challenges of IoT systems are also becoming more significant. One of them is system interoperability.

The diversity in IoT platform technology [4] and the networking protocols [5] have contributed to limited system interoperability in IoT applications. Numerous researchers have expressed device heterogeneity and interoperability as the two main issues that have been identified as the main obstacles of the progress of IoT [6,7,8,9,10,11,12]. Much has been done to solve the limitation of IoT devices interoperability with most of the proposed system architectures with fixed protocols, e.g., COMPOSE (Collaborative Open Market to Place Objects at your Service) [13] and Condense (reconfigurable knowledge acquisition systems) [14], or hardware operating systems, e.g., DIAT (Distributed Internet-like Architecture for Things) [15] and CEB (Cloud-Edge-Beneath) Atlas [16]. However, these different technologies compete to become the standard for IoT system. This causes difficulties to the standardization of IoT development and increases the requirements of extra hardware and software when connecting devices [17].

Moreover, most practitioners and engineers face extreme difficulties to meet the requirements of these architectures during real-world implementation. One typical reason is that different application domains have different design specifications, which require different technologies and face different development limitations. As a result, most of the real-world IoT systems leveraged model-based design methodology to build systems that solely suits the needs of the designated applications. These solutions might provide superior solutions for specific use cases but lead to a heterogeneous protocol landscape for IoT ecosystems and raise questions on how to enable a clear path for providers to utilize those IoT architectures [12]. Furthermore, large-scale implementation of these application-oriented-designed IoT solutions will require both network and domain knowledge and at the expert level to support the design or development of solutions for that specific domain [11,12]. On the other hand, the integration of uncorrelated sensors or IoT devices will be trivial while sometimes it could be extremely complex [18]. The introduction of Mobile Edge Computing (MEC) paradigm further extends the complexity of IoT system design due to the mobility management [19]. Hence, these explicit model-based solutions are not cost-effective due to the limited system and device interoperability, and low reusability.

Another problem is the ever-increasing data volume. Considering the growth in mobile systems such as mobile IoTs, WSNs, BSNs, robotics, unmanned aerial vehicles, and satellite systems, mobile data will challenge the storage and processing capacities of existing computing systems [20]. Many researchers have expressed concern towards the increase in data volume from IoT devices [6,21,22,23,24,25,26,27], and much data management solutions had been proposed. However, these systems’ mechanisms of managing and utilizing the data from these objects have yet to match the maturity of the IoT technology itself [21,28]. As data volume grows, the processes that depend on mankind to perform data analysis for knowledge extraction becomes inefficient and unrealistic. According to Cisco [29], five quintillion bytes of data are produced every day and it would take a lifetime to manually analyze the data process by a single sensor on a manufacturing assembly line. *Harvard Business Review* stated that only less than half of the structured data are actively used in decision-making and less than 1% of unstructured data is analyzed or used at all [29]. Hence, the process of converting data into useful information and knowledge still greatly relies on human effort. The data-driven approach is proven to be a solution to this issue. However, the currently available data-driven approaches are lack of flexibility and programmability to support interoperation for multiple application domains.

In short, the two highlighted issues are as follows:System interoperability: We define system interoperability for IoT systems as the ability of the system to accommodate and support the interoperation of hardware devices with different operating systems (OS) and protocols seamlessly. Since existing IoT devices operates under diverse OS and protocols, the creation of a IoT platform architecture that is flexible to integrate with these devices is a challenge.Lack of flexibility in data-driven approach: The increase in data volume that is beyond possible for human effort to extract information reduces the usability of IoT systems. The data-driven approach is proven to be a solution to this issue. However, the currently available data-driven approaches lack flexibility and programmability to support the interoperation of application domains.

To solve the issues mentioned above, we propose a loosely-coupled, component-based architecture for IoT that enables data-driven automation. These are our key contributions:A component-based IoT architecture that supports system interoperation between IoT devices with different OS: The proposed architecture treats each component as an individual service provider that operates independent of other services components. This enables the accommodation of model-based IoT systems under one architecture with only the least required interface between each system.The realization of flexible data-driven feedback control for IoT architecture: A proof-of-concept IoT system with the mechanism to perform data-driven automation is showcased. Furthermore, the easiness of data-driven optimization model interchange and update is demonstrated.

The rest of the article is organized as follows: Section 2 briefly discusses some related work on IoT architectural design principals. In Section 3, we describe and explain our proposed IoT architecture in detail. The description and implementation of the composite real-world use case scenarios that we selected to apply and test the prototype IoT architecture are discussed in Section 4. In Section 5, the experiment results and findings are presented. Finally, we conclude the paper in Section 6.

## 2. Related Work

We review some of the essential characteristics of a data-driven IoT architecture, aiming to solve heterogeneity, interoperability and reusability issues. Besides the basic functionalities, we narrow down some of the key aspects that are essential for the development of high interoperable, data-driven IoT architecture. A brief explanation of each characteristic is as follows:System interoperability: The ability to accommodate devices of diverse operating systems or protocols. The system development method and choices of protocols or devices are diversified depending on application domains. As a result, this created the interoperability issue for IoT systems. To realize device interoperability, the system needs to comply with a certain standard of system architecture, such as operating systems or protocols.System reusability: The ability of the system to be reused for any identical or different application. The system reusability is essential for wide-scale adoption of IoT in the real world as the ease of system duplication greatly reduces the effort required to design, develop and deploy an IoT system.Data-driven function availability: The existence of any decision-making or action mechanism based on the data sensed. Data-driven modeling (DDM) is based on analyzing the data about a system, in particular, to find connections between the system state variables, such as input and output variables, without explicit knowledge of the physical behavior of the system [30]. The implementation could be a heuristic closed-loop feedback mechanism or machine learning and artificial intelligence techniques.

Based on these characteristics, we review a few data-driven IoT architectures which have targeted to solve heterogeneity and interoperability issues. For instance, Distributed Internet-like Architecture for Things (DIAT) was introduced by Sarkar et al. [15], as a solution for large scale expansion of IoT. The proposed idea specifically created a standardized operating system (IoT daemon) with layered architecture that provides various levels of abstraction. The devices under this architecture will have different layers of the same OS according to the services they provide and the resources they have. The advantages of having a standardized OS includes faster and seamless device interfacing and information processing. Furthermore, it is easier to perform system maintenance and updates for the entire system that operates under the same OS. However, the proposed idea failed to address device heterogeneity and scalability issues as the system will not support devices of different OS. Similarly, Xu and Helal introduced Cloud-Edge-Beneath (CEB) Atlas architecture [16]. The architecture proposed a standard description language that fosters the interoperation and integration of the IoT systems. However, the architecture has limited interoperability, as the proposed architecture only works on dedicated Atlas platform, which is a smart device platform developed by the same group of authors. These types of IoT architectures target to provide solutions for generic application but mostly suggested their own software or hardware protocol which restrict heterogeneity and interoperability.

Different from the mentioned architectures, Calderoni et al. [18] identified the need for IoT architectures to provide a here degree of interoperability through minimal standardization of system protocol and introduced the IoT Manager. The proposed solution aims to provide a highly interoperable, full-stack IoT platforms that enable rapid deployment of IoT prototypes, through the quick coupling and decoupling of sensors network, even for those which are already online. One key feature of the IoT Manager is to be able to accommodate a variety of sensor devices as long as they communicate under HTTP or HTTPS-based API protocols. Balasubramanian et al. [31] proposed a MEC-based architecture for energy harvesting IoT devices integration, known as the 2EA (Energy-Aware-Edge-Aware) architecture that mainly tackle two problems: (1) the offloading of data traffic from IoT devices; and (2) resources assignment at the Mobile Edge Computing system. The proposal comes with a data-driven model for MEC computation and network resources allocation. Similarly, Guimarães et al. [12] proposed the FIFu (Future Internet Fusion), a network architecture that unifies existing and upcoming network architecture. The solution target to resolve heterogeneity that exists upon currently available network architectures and protocols through the identification of network protocols and conversion of messages. Benayache et al. [32] suggested a middleware architecture known as the Microservice Middleware (MsM) to tackle device heterogeneity and interoperability issues. An Artificial Neural Network-inspired concept is used to build the standard microservice architecture for IoT device classification and integration. The proposed idea has also adopted the loosely-coupled design method to support device heterogeneity but the system interoperability and data-driven functions were not covered. Li et al. [11] proposed IoT-CANE, a unified configuration management and recommendation system for IoT, cloud and edge infrastructure. The proposal aims to solve the heterogeneity and complexity of cloud/edge/IoT environment by implementing a system that can be updated when IoT solutions evolve to facilitate increased data and knowledge acquisition. It is a different approach which does not provide a generic IoT architecture but proposed a comprehensive IoT resources and services classifier to enable easier integration of new services to the existing IoT systems.

Another major challenge during the realization of system interoperation is to enable network interface between IoT devices with different networking protocols. To tackle this problem, the Condense architecture is introduced by Vukobratovic et al. [14] as a solution for interoperability and scalability issue of IoT systems. The main focuses are to provide an active and reconfigurable service leveraged by the data analysis process and to enable the integration of Condense into 3GPP (3rd Generation Partnership Project) machine type communication architecture. However, the implementation only focused on the network resources allocation, which does not solve the interoperability issue for a system of various application domains [21].

With most IoT solutions designed and developed using various tools and platforms, it is unrealistic to select one as the standard. Furthermore, architectures that have their own standard device platforms or protocols is not an effective method to solve interoperability issue. To realize system interoperability and reusability, the requirement of IoT architectures to operate under a standard OS or protocol should not be removed. Instead of promoting interoperability by using one standard protocol, a method that interconnects each different model in a loosely-coupled manner would be a better solution. The combination of such multiple models is able to contribute a more comprehensive solution as they add up in the right measures and sequences to provide a complete solution to users [15].

Another concern is the availability of the data-driven functionalities as they are essential to compensate for the drastic increase in data volume. IoT systems that provide cognitive data-driven functionalities are receiving more attention than ever before. For instance, CEB and DIAT are both integrated with cognitive decision-making model that operates based on predefined situations. Unlike generic architecture, such as CEB and DIAT, which build the cognitive functions for system optimization, domain-specific architectures build data-driven models for domain related optimization. For instance, Predescu, Mocanu, and Lupu [33] developed a real-time IoT control system for pumping stations of a water distribution system. The proposed architecture aimed to solve the problem of water flow and pressure control of a water distribution network with respect to the demand and possible leakage scenario. These data-driven cognitive functions have increased the effectiveness of the system.

Table 1 compares all the IoT architectures based-on the characteristics mentioned above. By using the number of “Yes” answers given to each architecture as an evaluation metric, we can see that DIAT, CEB Atlas, MsM and IoT manager have equally scored two “Yes” answers. Hence, a qualitative comparison was conducted using our proposed architecture against these architectures. Overall, we identified the gap of currently available IoT architecture, which is low interoperability and the lack of data-driven functionalities. Therefore, we intend to provide an IoT architecture that tackles interoperability and reusability limitations while focusing more on the application and service optimization through an easily interchangeable and programmable data-driven cognitive functions development platform.

## 3. Component-Based IoT Architecture

The illustration of the IoT architecture is shown in Figure 1. Through the adoption of the component-based design method, we defined five major types of classification for devices and services that interact under the proposed architecture: (i) central control system component; (ii) end devices component; (iii) data-driven feedback model component; (iv) external data input component; and (v) software application component. The abstract representations of these classified components are provided to efficiently create and manage the services of each component. Based on the classification, the interface protocols and ways of interacting between each component are determined. The design of the architecture adopted mobile edge computing (MEC) technology that greatly reduces the latency by placing the computational power at the edge to meet the real-time needs of IoT applications [35].

To perform specific tasks independently or corresponding to other components, the components need to expose their services in a homogeneous way. Hence, the proposed IoT architecture should be able to accommodate arbitrary components and their operating behaviors, while standardizing the interconnection protocol for each type of components, especially dealing with low-level hardware. We intend to design an IoT platform architecture that is able to support and accommodate the interoperations of devices operating under different OS. Unlike systems that require all the devices to have the same OS for easier system maintenance and management, our proposal focuses on enabling quick and easy “plug-and-play” of devices or IoT systems for the instant prototyping with only the data format and communication protocol to be standardized.

### 3.1. Central Control System Component

The central control system component hosts the central platform and server that manage all the interactions between components. The component is set to provide only general functions such as data or command interchange, data process, and connectivity management of the components to enhance the system’s interoperability and reusability. There are five sub-components under this component:Data string decoder: Data string decoder is a sub-component that decodes the data string received from the end devices into a standard, readable-data format. A standardized data format that aggregates all the substantial values from one device is required to prevent data mismatch between devices, considering the possibility of a single device providing more than one type of data. The data format from the end devices generated by the data string generator in the end devices component is explained in the next section. Typically, the data from the end devices have to be processed into a standardized and readable format before being transferred to the data-driven feedback model component for further analysis and the database server for storage. Therefore, the data string decoder identifies the separators between essential data values in a data string and extracts the required data, which contain the sensor’s data and the device ID. The extracted data and ID are then be sent to the device identification manager for identification and classification.Device identification manager: This is the sub-component that identifies and classifies the ID and sensors’ data of the end devices and matches them with their corresponding actions. Firstly, the sub-component fetches the device ID from the data string decoder and tries to find a match in the existing device ID registry. If a match is found, the data under the device ID are transferred to the data-driven model selector for further action and also to their corresponding database for data storage simultaneously. The function also automatically creates a new ID for a device if the device ID is not to be found in the repository. The identification of devices is based on their multi-level device ID which enables the segregation of data from the application level down to the device level. The detail explanation of the multi-level naming system for end devices is covered in the next section.Device connectivity manager: This sub-component handles the connectivity and traffics between each component. The most important function of this sub-component is to maintain the parallel and simultaneous data transmission of end devices and external data sources, with the central control system component. This is to ensure that the data from each end device do not have to wait for their turns to be read. Moreover, the sub-component also ensures that the central control system component only reads a new data string from the end device when the feedback generated by the data-driven model from the previous inbound data has been successfully sent to the corresponding end device. This is to make sure that the feedback signal received by each end device is the feedback generated from the data that they have provided during the last iteration. The interface and communication of this component with the other components are achieved through general, open-sourced protocols: API call and MQTT. Specifically, the usage of MQTT protocols on top of Wi-Fi TCP/IP is suitable for the end device communications as MQTT requires only one single IP address for all the devices connected under the same gateway. This is appealing for the large-scale deployment of this architecture as it can accommodate a larger network of devices. By leveraging the edge devices as a gateway connecting multiple end devices, the end devices are able to exchange data and interact through the single IP address of the edge gateway that they are connected to. In addition, the MQTT protocol’s multi-level topic assignment is suitable for the implementation of the multi-level naming system. The topic and message structure of the MQTT protocol enables devices to receive data only from the designated party(s) based on the topic and level of subscription.External data handler: This sub-component is in-charge of passing the data from the external data input component to the software application components for visualization and data storage. The main functions include external data source identification, database creation, and sensor data logging.Device ID registry: The device ID registry is a data registry that stores all the IDs of sensors and external data sources that have previously been processed by the system. To effectively support multiple applications under one IoT system, the registry stores the device IDs and external data sources according to their applications and data types. It is used by the device identification manager to cross-reference against the incoming device ID from the end device component. The data of this registry are stored within the database of the system, together with all the sensors and external data for instant referencing.

### 3.2. End Devices Component

The end device component represents all the lowest-level hardware and consists of sensor nodes such as microcontrollers and microprocessors that integrates with sensors or actuators. There are two major sub-components under this component:Data string generator: The data string generator is a function that aggregates the data from a single sensor into a standardized data string format. The sensor data collected are concatenated into one string message to ensure the data from the same device is transferred and process exactly at the same time. We define the string format of the data from the end devices to be as follows: “*Data 1*/*Data 2*/…”, with solidus (/) sign as the separators of each reading. This is especially useful during the transmission of time-sensitive sensor values, as it reduces the possibility of data mismatch when more than one time-based data are transmitted at the same time. As most microcontrollers operate under different frequency and clockwise, the data sensing operation of more than one sensor under one sensor node will create a time discrepancy between different sensors, depending on which sensor is programmed to sense or read first.Multi-level naming system: The multi-level naming system for the end devices under this component is used for the end device ID creation. To support device interoperation, the devices and data should be well segregated despite being stored and processed under the same server. The naming of end devices begins with the categorization based on application domain followed by the services provided by them: *“APPLICATION/SERVICE/#”*, where “#” represents the number order of devices. An example is: *“E_HEALTH/TEMPERATURE/1”*. The ID of the end devices is designed to be readable for easy identification and modification. The level definition of each naming of information is essential for data labeling, especially when the node device provides more than one type of sensor readings.

### 3.3. Data-driven Feedback Model Component

The data-driven feedback model component is a Unix-like development environment for the data-driven models for smart optimization and prediction. Users can build new data-driven models, select or update existing data-driven models in this component. Considering the latency and data integrity concern for an analysis and decision-making component, we have built the component on top of the edge device, where it operates on the same device as the central control system component. It greatly reduces the latency of data transmission and the possibility of data loss in the process of data transferring between components. There are two main sub-components under this component:Data-driven models: This sub-component stores all the data-driven models. The component is built as an entirely independent component to ensure the changes in the models will never affect the operations of other components. Furthermore, since both the data-driven models in this sub-component and the data from the central control system component are declared as global variables, they can be shared across different application domains or optimization models for more comprehensive optimization results. For instance, a linear regression model can be used to predict future household electricity consumption, water consumption and other applications under one unified system. Similarly, the data input of a solar irradiance sensor used for solar intensity prediction can be used by the weather prediction model or any application that makes the variable call. In short, these models can be utilized in more than one application and data can be processed by more than one data-driven models to provide higher system flexibility and interoperability.Data-driven model selector: This sub-component is an automated model selector and assigner for the end devices. Users can conduct performance evaluations on each algorithm, under the same, specific model or situation. Through the evaluation, users can sort out the optimum algorithm and rank the performance of different models tested under the same parameters. In addition, practitioners can also validate and test newly developed optimization or prediction models under the deployed system, which provides a realistic environment for model validation. Algorithm 1 describes the data-driven selection mechanism of the data-driven model selector. The algorithm first extracts the device ID into *deviceID* and sensor value into *sensorValue* from the in-feed data of each end device. After that, based on the preset data-driven model assigned to each *deviceID*, the data are processed and feedback is generated to update the operating parameter of the end device. The generated feedback is then stored in the database for reference and performance comparison other models used by the same *deviceID* previously. If there is no model from the *modelList* pre-assigned to the *deviceID*, the data are processed by the first model in the *modelList* until convergence and all the generated feedback data are stored to the database. During the next iteration of optimization, the second model is used to generate feedback until convergence and this continues until all the models in the *modelList* are used. The stored data of all the models are compared against each other to select the model with the best optimization performance for that specific *deviceID*. Lastly, the model id assigned to the *deviceID* for future feedback generations.
**Algorithm 1** Data-Driven Model Selector1:  **for** each end device **do**2:    extract deviceID3:    extract sensorValue4:    **for** each *i*∈[0, length(modelList)−1] **do**5:        **if**
deviceID = modelList[i] **then**6:           modelList[i]←sensorValue7:           generate feedback8:        **else if**
prevModel = modelList[i] **then**9:           modelList[i+1]←sensorValue10:           generate feedback11:        **else**12:           modelList[0]←sensorValue13:           generate feedback14:        **end if**15:        update operating parameter of end device with feedback16:        store feedback in database17:    **end for**18:  **end for**


### 3.4. External Data Input Component

External data input component represents any reference data that is not directly read from the end devices, for example API data from weather forecast station, solar activities’ data from an open-source website or smart devices computational data. In this component, we have two sub-components: (i) web API data extractor; and (ii) data extractor scheduler.
Web API data extractor: A web API data extractor is a sub-component to curl the data from any designated API. It also includes decode of the data downloaded through web API into JSON format through a curl script.API data extractor scheduler: The API data extractor scheduler is a function that automatically repeats the data extraction process by a fixed interval set by the user to receive the latest data from the API data provider.

After the data processing, the essential data are transferred and stored in the same server as the other data from the end devices component. These external data are also visualized under the same platform as the other data through the software application component.

### 3.5. Software Application Component

The software application component consists of two main sub-components: (i) configurable data visualization platform; and (ii) database management system:Configurable data visualization platform: The configurable data visualization platform is a data visualization web application and graphical user interface (GUI) for data monitoring panel controls. To enable system interoperation under one platform and reusable for different application, the data visualization module provides a modifiable data presentation platform to suit the requirement of multiple application domains. Furthermore, considering the wide adaption and usage of Business Intelligence (BI) data analytic tools such as Microsoft’s Power BI and SAP Business Intelligence, customizable and flexible data visualization platform is useful for a more comprehensive data presentations. However, it is also crucial to reduce the complexity of data pre-processes to increase the platform’s usability. Therefore, this component aims to maximize the customizability of the data visualization platform and to minimize the requirement of complicated data pre-processes by providing a customizable and flexible data visualization platform that can directly interface with the databases of the IoT systems.Database management system: The database management system is a generic database system with an essential feature: automatic database generation for new incoming data. The central control system identifies the databases and data-driven models for incoming data based on their device ID and determine whether the database for the data from this device has already been created.

Overall, the heterogeneous data are presentable on one single platform for easier data analysis and information extraction.

## 4. Use Case and Implementation

### 4.1. Use Case: Solar PV Maximum Power Point Tracker (MPPT)

The Maximum Power Point Tracking (MPPT) is a technique commonly used in solar photovoltaic (PV) systems to locate and maintain the highest operating power point of the solar PV. Solar PV cells have a complex relationship between solar intensity and operating resistance across the cells which produces a non-linear output efficiency that can be analyzed based on the *I-V* curve. This non-linear property creates a phenomenon where only a specific operating voltage point can contribute to the maximum power point. Therefore, maintaining the solar PV system to operate at its maximum power point is vital to ensure the maximum solar energy extraction of the system [36].

Most of the MPPT techniques control and adjust the operating voltage or current values of the PV modules to achieve maximum power point. The MPPT controller’s algorithm first examines whether the solar PV module is operating at its Maximum Power Point (MPP). If the PV module is not operating at its MPP, the algorithm will calculate and locate the MPP of the solar PV. After that, the MPPT controller will send a command signal to the converter to make adjustments on its duty cycle (operating voltage). There are various MPPT algorithms available and two of the most widely applied algorithms are Perturb and Observe (P&O) [37] and Incremental Conductance (InCond) [38] algorithms. Both algorithms adopted the “hill-climbing” concept, which operates by initially measuring the voltage or current data to identify the current power point of the solar PV. The system then moves the operation point (voltage or current) in the direction where the power calculated increases. The increment continues until a decrease in power is detected, and the next perturbation is in in the opposite direction. The perturbation continues to oscillate around the MPP as long as there is no power drop.

We adopted solar MPPT as our use-case for the following reasons: (1) Most solar MPPT devices are offline, mechanical devices that operate individually at their local area. The high complexity in MPPT technology and the diversity in MPPT techniques made MPPT devices difficult to be put online for centralized performance monitoring. Hence, it is a suitable use case to showcase the system interoperability limitation. (2) Being a highly time-sensitive system that requires great precision and instant response, the MPPT system is a suitable use case to test our proposed architecture on the suitability and viability to be applied in real-world scenarios and the realization of data-driven feedback mechanism.

### 4.2. Implementation

To validate the feasibility of the proposed architecture, we built a prototype edge IoT system and deployed the system to the real-world for experimental purpose.

The hardware architecture illustration of the proposed edge IoT system is shown in Figure 2. Firstly, Raspberry Pi 3 is used as the edge device to host the central control system. The data transmission from the edge device to the data storage server is also achieved through the MQTT protocol. Arduino MKR1000 microcontroller (Arduino, Massachusetts, USA) is used as the hardware for end device components.

We integrated the MKR1000 microcontroller with a voltage and a current sensor to detect the operating voltage and current of the solar PVs. The microcontroller wirelessly transmits the sensed data through the MQTT protocol to the Raspberry Pi 3. The Raspberry Pi 3 then processes the inbound data and generates a feedback value to the MKR1000 microcontroller. The microcontroller receives the feedback signal, converts the received feedback into a PWM signal and apply the signal by matching it with the operating frequency of the solar PV power converter to update the operating voltage of the solar PV.

Finally, the data are streamed to the database server for storage. The data transmitted from the end devices are transferred to a server through MQTT, from the edge device as soon as the data is received. Since our data are time series data, we decided to use *InfluxDB* [39], which is a time series database (TSDB). It is optimized for fast storage and retrieval of time series data for application such as operations monitoring, application metrics, IoT sensor data, and real-time analytics. Apart from that, *InfluxDB* is integrated with MQTT plugins, which enables direct MQTT messages storage in the database without MQTT broker installation or any extra effort to process the data. Furthermore, the data in *InfluxDB* can be easily visualized through *Grafana* [40], which is a customizable data visualization platform that supports direct integration with *InfluxDB*. The data from the external data input component are also streamed to the same server.

The overall data-driven IoT system architecture layer illustration is shown in Figure 3. The main functions consist of data processing and data-driven feedback functions. The data processing function receives the inbound data from microcontrollers in string format and decodes the message to extract the necessary sensor values. The function recognizes the device ID through the subscribed MQTT topics. For instance, we programmed the first solar PV’s end device to publish the data string under the topic “Sensor/PV_1” and the second solar PV’s end device as “Sensor/PV_2”. The data string is processed to extract the voltage and current readings. After processing the data, the function transmits the extracted sensor values to the data-driven feedback model and data storage server simultaneously. The program flow of the overall program of the proposed IoT system is shown in Figure 4. The explanations for each function are as follows.Data Processing Function: Firstly, the system receives data from the voltage and current sensors, and the sensor IDs are identified through the topic of the MQTT messages. After that, the message, which is in the format “*Voltage*/*Current*”, is decoded to extract voltage and current values. The extracted data are then labeled according to their respective device ID and are assigned to the corresponding data-driven feedback model. For instance, the voltage and current data with device ID “*PV*_#” are assigned to their respective MPPT data-driven feedback model for feedback generation.Data-driven model: The data-driven model generates feedback through the MPPT algorithms: Perturb and Observe (P&O) and Incremental Conductance (InCond). Users can select or update the algorithms without the need to manually reassign the data to the newly-selected model or to change the data input format, as the data-driven models react to the same set of input data. Furthermore, under the same data-driven model, the assignment of the different algorithm can be done. For instance, user can assign P&O algorithm for PV_1 and InCond algorithm for PV_2, and vice versa. After going through the MPPT calculation, the feedback signals are transmitted back to the end devices to be applied on the solar PV’s DC-DC converter to update the controller’s action. The program then continues its iteration to read the next data set to decide the upcoming feedback action.External data input handler: The external data input handler is in charge of data curling from any website on the Internet through API. For our use case, we import external weather data from the *OpenWeatherMap* as a reference. The *OpenWeatherMap* is an online service that provides weather data, such as daily weather forecast and historical data. The data provided are available in JSON format and accessible through free API. However, only certain data (current weather data and five-day/3-h forecast) are available for free. For experiment purpose, we only collected these data, stored them in *InfluxDB* on a daily basis, and visualized the data on *Grafana*.

## 5. Experiment Results and Discussions

Firstly, an experiment was conducted to test the data-driven feedback mechanism’s operation and the viability of the system to be implemented on a real-world use case, which is the MPPT application. After that, another experiment was conducted to examine the ability of our system to support system interoperability, reusability and device heterogeneity.

### 5.1. MPPT Implementation Experiment

The experimental setup is as shown in Figure 5a,b, which is an experiment setup adopted from [36]. A solar array simulator (SAS) from Chroma (Model: 62150H-1000S) was used to generate the output characteristics of a solar PV array. The Chroma SAS is a DC power source that is capable of simulating the *I-V* curve of different solar PV arrays under various solar intensity. For our case, a six series-connected PV modules of Kyocera Solar (Model: KC85T) was used to build the PV array in the SAS, and PV module specifications are shown in Table 2.

The solar intensity and solar module model were selected from the SAS software. A single-ended primary-inductor converter (SEPIC) with 10 Ω resistor load was used as the converter to step-up or step-down the operating voltage of the solar PV. The input of the converter was connected to the SAS, which acted as a solar array output. The voltage and current values of the PV array were sensed by the LEM sensors (LV-25P and LA-25NP), which were used as a voltage and current transducer in this experimental setup. The sampling rate of the sensors to measure the voltage and current from the system was 0.1 s. The MKR1000, which acted as an end device of our IoT system, was connected to the sensory circuitry to read the PV array’s operating voltage and current while integrated to the SEPIC converter. The Arduino MKR1000 was integrated with the Raspberry Pi 3, which was wirelessly connected to the Internet, and, whenever the MKR1000 measured a set of voltage and current reading, it transferred the data to the edge device, and waited for the feedback from the edge, as the next reading was only measured after the feedback received was applied on SEPIC. The Raspberry Pi as our edge device collected and processed the data from MKR1000 to calculate the operating power point. According to the power point calculated, the new duty cycle of the SEPIC was generated, and feedback to the MKR1000 was applied to the SEPIC. This operation automatically repeated until the maximum power point was located. The duty cycle step size was ±0.01 and the switching frequency was 25 kHz. At the same time, the raw data of voltage and power collected from the end device were sent to the *InfluxDB* for data visualization through *Grafana*.

The proposed approach was evaluated for two different solar intensity settings: (1) 334 W/m${}^{2}$; and (2) 667 W/m${}^{2}$. Firstly, the P&O algorithm was used in both experimental setups. The experiment was then repeated with InCond algorithm. The sampling time to measure the voltage and current of the system was fixed at 0.1 s. First, we successfully showcase that the change of the MPPT algorithm is easily achievable in Figure 6a,b, where we change the MPPT algorithm and conducted the experiment without changing the experiment setup, data format, and end devices’ settings. Figure 7 and Figure 8 show the *I-V* and *P-V* curves both solar intensities, where the red dots on the curves indicate the maximum power point (MPP) of the solar array simulators. The MPPT tracking efficiency percentage values are highlighted with red-colored boxes. We can see that the system accurately tracked the MPP with MPPT efficiencies of above 99% (99.08% for 334 W/m${}^{2}$ setup and 99.80% for 667 W/m${}^{2}$ setup).

Furthermore, we can see that our system is able to track the maximum power point and maintain the operating parameters at the MPP. Figure 9a,b shows the tracking results of both solar intensity settings. Observing the tracking results, the x-axis scale, which is 0.5 s per box in all figures, on the left-side shows that the system accurately tracked the MPP within 2 s and maintained the MPP continuously. This proved that the proposed system is able to perform data-driven feedback successfully and able to deliver the performance that exactly the same as existing offline MPPT controllers. This also indicates that the proposed systems response speed is high enough to support time-sensitive system. In addition, the transmissions of data to the *InfluxDB* and the data visualization through *Grafana* is also successfully showcased in Figure 10a,b. Overall, the end-to-end functions of a data-driven IoT architecture are shown through the conducted experiments.

### 5.2. System Interoperation Experiment

The experiment setup is shown in Figure 11. We used a PC as a MQTT broker to send data to our edge device. We also used an Arduino Uno and a MKR1000 to simultaneously transfer their respective data, with the same format as mentioned in Section 3, to the edge device of our IoT system. The experiment results were obtained through the timestamp of each data logged into the *InfluxDB*. As shown in Figure 12, the Raspberry Pi 3 is able to receive data, generate and transmit data-driven feedback, and stream data into a storage server, as soon as the data are received, with zero response delay.

Observing Figure 13, we can see that the data from all three end devices are displayed under the same time interval. Hence, we verified that the data streamed to the *InfluxDB* are stored instantaneously. Through the experiment, we verified that the IoT system is able to accommodate system interoperation and support device heterogeneity by integration with devices that runs on different operating systems. The receiving and transmission of data, and data-driven feedback provision are achievable by the IoT system simultaneously with no delay. Furthermore, we have showcased that the proposed architecture can easily support the integration and plug-and-play of IoT systems with other devices of a different OS without complicated amendments on both the main IoT system architecture and the newly integrated devices. As a result, the loosely-coupled nature of the proposed architecture provides a high degree of reusability as existing IoT system can be reused without much modification.

### 5.3. Discussions

After conducting the experiments and successfully verifying the feasibility and viability of the proposed IoT architecture, a comparison of the proposed architecture with the four reviewed architectures that have the highest score in Section 2—DIAT, CEB Atlas, MsM, and IoT manager—was performed. We evaluates them based on two main characteristics: (1) the support for device heterogeneity; and (2) the flexibility of data-driven function.

### 5.4. System Interoperability

The proposed IoT architecture focuses on providing a solution to support the system interoperation, reusability and instant prototyping of IoT system through a component-based approach. The loosely-coupled integration of devices enables the interaction with minimal standardization in terms of operating system and connectivity protocol. In addition, the data-driven model selector provides real-world evaluated model recommendation and high programmability for greater performance optimization for IoT applications.

The DIAT architecture, however, requires the IoT daemon of the architecture to be installed in all the devices integrated into the architecture. Furthermore, despite having the flexibility to install only the required layer of the IoT daemon depending on the devices’ functionalities in the system, these IoT devices are still restricted to operate with the devices or systems with the same IoT daemon installed. Moreover, DIAT does not assume that IoT systems’ owner and manufacturers are driven by different interests and binds user with a specific manufacturer, which does not provide all the devices or services one needs [41]. Similarly, when compared with the CEB Atlas, our proposed architecture provides a higher degree of system flexibility and device heterogeneity support as the CEB Atlas requires the IoT architecture to utilize devices operating on the Atlas platform, whereas our proposed architecture can support devices with a variety of operating OS. As for MsM and IoT manager, they provided a similar level of requirement for communication protocol as our proposed architecture. However, both the MsM and IoT manager lacks data-driven modules.

The architecture comparison is shown in Figure 14, which visualizes the IoT architectures based on the basic components that are required for the devices to interact with each other, which includes a standardized OS (IoT daemon) and a standardized connectivity protocol. In the figure, we can observe that our proposed IoT architecture is the same as MsM and IoT manager, which require only one component to be standardized between devices, whereas the DIAT and CEB Atlas architecture require at least two components to be standardized between devices. As a result, our proposed architecture provides the highest degree of system flexibility and support for device heterogeneity and enables the easiest device and system interoperability and reusability. Different from the MsM and IoT Manager, our proposal is equipped with a data-driven module, which is further discussed in the next subsection.

### 5.5. The Flexibility of Data-Driven Function

Comparing with the event-driven optimization model of the DIAT architecture and the E-SODA of the CEB Atlas architecture, the data-driven feedback model component of our proposed architecture provides a greater degree of flexibility for users to customize the algorithms according to the need of users, as the data-driven function is developed as a standalone component. It can operate independently by only receiving data processed by the central control system component. The comparison of the proposed IoT architecture, DIAT and CEB Atlas architecture is shown in Figure 15. In the figure, we can observe that our proposed architecture have a loosely-coupled data processing function and data-driven models development environment. It is different from the data-driven models of the other two architectures, which have combined the data processing function and the optimization algorithm. One of the advantages of the component-based IoT architecture proposed is the flexibility of each component to be attached or detached, without affecting the operation of the remaining components. As a result, the data-driven functions are flexible to be updated, customized or altered easily. Moreover, the data-driven model selector embedded in the system serves as an optimization model evaluator and recommender that provide higher efficiency and performance yield for real-world application.

### 5.6. Potential Application Domains and Use-Cases

The application of the proposed IoT system as a MPPT tracker has proved the feasibility of the proposal to be utilized for the real-world scenario, targets to support most IoT system applications, as long as they operate in a “data collection -> stochastic analysis -> feedback and visualization” manner. Moreover, the proposed architecture aimed to support large sensors and controllers network of same or different OS. Hence, use cases that use IoT system to interact with a huge number of end devices are suitable to be implemented using the proposed architecture. We suggest three use-cases that are suitable to be implemented using the proposed architecture:Smart factory: Smart factory systems generally interact with a large network of sensors and controllers that requires real-time data sensing and feedback action. Furthermore, these end devices are typically operating under diverse protocols such as RFID, Modbus, and MQTT.Household smart grid: Household smart grid typically manages both the supply and demand of electricity of a household. The supply-side consists of the electricity from the national grid and any electricity generated from renewable energy, whereas the demand-side consists of the appliances that consume electricity. In short, a household smart grid system is required to manage sensors and relays of both supply and demand-side, which are two different systems integrated on one platform. For instance, a smart grid system is deemed to provide instant adjustment on household electricity generation during high solar irradiance; or electricity consumption during on/off-peak period throughout the day, or when the user is not within the premise.Smart city: Smart city systems are combinations of multiple IoT applications including smart grids, smart traffic systems, and smart home systems with highly condense network of sensors, CCTVs, and controllers. In fact, the smart city is one of the most ideal use cases that is suitable to be implemented using the proposed architecture due to its cross-domain integration and high-density data volume that requires data-driven feedback models.

However, the proven interoperability and reusability of the proposed architecture may be compromised under different environmental conditions such as drastic weather and geographical conditions, limited network resources and computational resources. Hence, the proposed architecture is not suitable to be implemented in extreme environments such as oil and gas extraction and refinement stations that are usually located in the middle of oceans with low network availability. Another concern of the proposed architecture is the lack of support for distributed machine learning or optimization application. Despite being loosely-coupled and highly distributed, the data-driven models are still stored and operate under one machine which raised concern on the limited computational resources. As a result, the proposed architecture is unable to support a domain that uses highly-complex or distributed optimization models, such as Federated machine learning [42] or distributed machine learning [43]. The possibility to achieve distributed and parallel optimization remain as a challenge for future research.

## 6. Conclusions

This paper presents a component-based architecture for IoT systems that supports system interoperability and service reusability, while able to accommodate flexible data-driven feedback functions. We validated the viability of the proposed architecture by building a prototype system based on the proposed IoT architecture.

We implemented the prototype IoT system on real-world use case scenarios, which is the Solar PV MPPT system, and conducted a series of experiments: (1) MPPT implementation experiment; and (2) system interoperability experiment. Through Experiment (1), we proved that our proposed architecture is viable to be implemented in real-world situations. The data-driven functionality is demonstrated through the implementation of the MPPT algorithms. Furthermore, it is able to deliver the same performance as existing offline MPPT systems.

Through Experiment (2), we managed to show that our system is able to support the interoperability of different devices. The system can interoperate with different kinds of devices. In short, the system demonstrated a higher level of support for heterogeneous system and device interoperations with minimum standardization required. Moreover, the reusability of components is also demonstrated through the experiments, as we are able to easily integrate multiple devices of different OS without needing to alter the main system framework and other components of the system.

Overall, our proposed idea is proven to be able to solve the system interoperability and reusability limitation effectively through the implementation of IoT system prototype in a real-world use case. Furthermore, we introduced the automated data-driven function approach, which increases the cognitivity of the IoT system on data analysis through the automated data-driven model selector that self-evaluates and chooses the best performing model for specific applications. The component-based design method is proven to be able to accommodate system reusability and interoperability. The current minimum standardization required for our system to realize interoperation is the communication protocol coupled with a fixed data format. To realize a greater degree of interoperability, the future research direction is to further remove the standard requirements on the data format. Since the basic communication between IoT devices that operates under different communication protocols is currently available, a cognitive system that is able to resolve data string of diverse communication protocols and a smart data decoding and identification system for non-standardized data formats are required to further enhance the system interoperability of IoT systems, which is highly achievable through machine learning and AI methods.

## Figures and Tables

**Figure 1 sensors-19-04354-f001:**
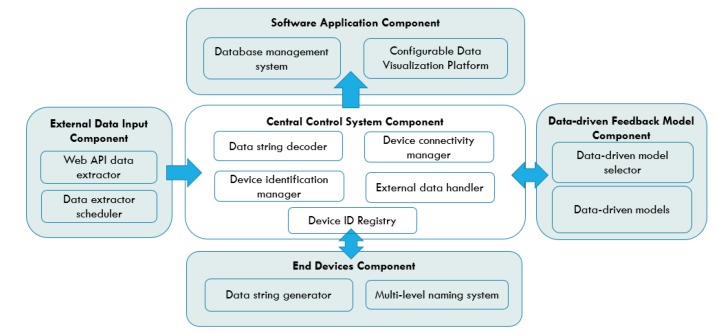
The components of the proposed IoT architecture.

**Figure 2 sensors-19-04354-f002:**
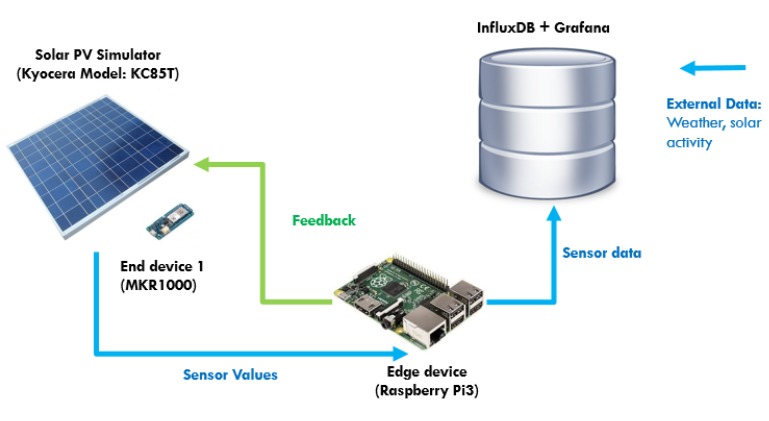
Hardware illustration of Edge IoT system (MPPT use case).

**Figure 3 sensors-19-04354-f003:**
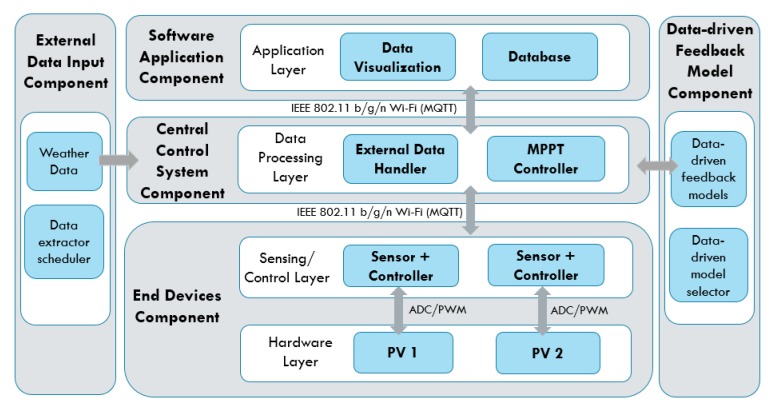
System architecture layer illustration (MPPT use case).

**Figure 4 sensors-19-04354-f004:**
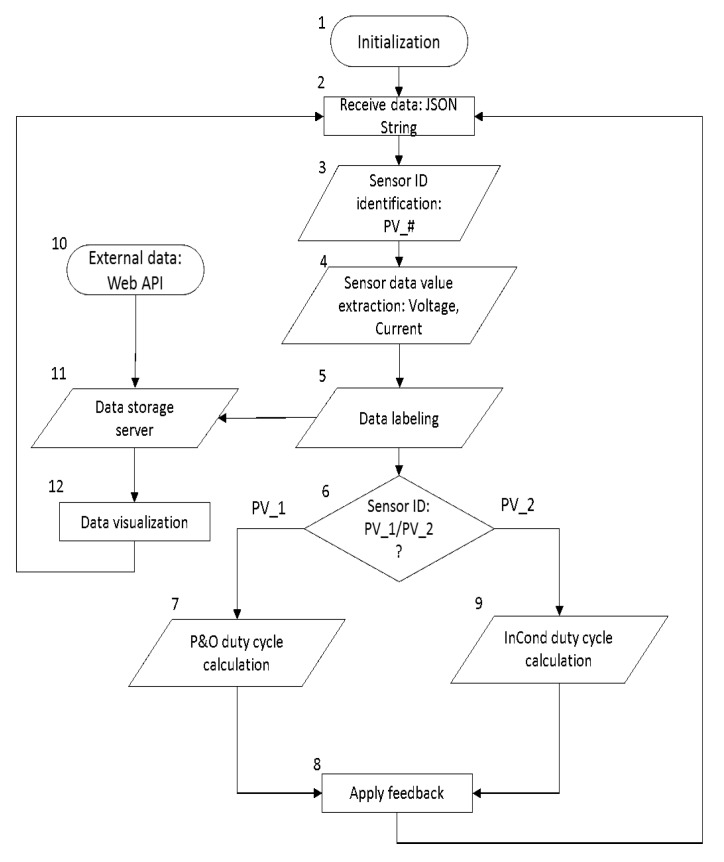
Program flow chart of the data processing and data-driven function (MPPT use case).

**Figure 5 sensors-19-04354-f005:**
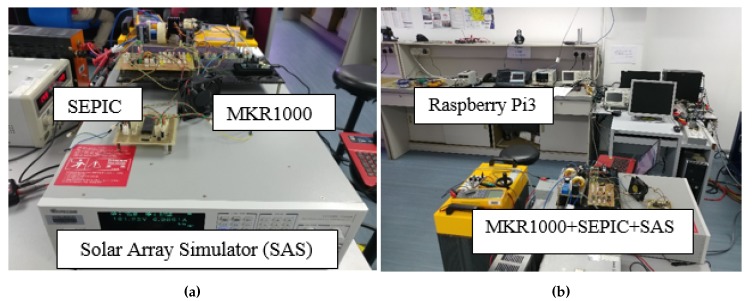
Experiment setup of the MPPT implementation experiment. (**a**) end device setup, (**b**) overall setup.

**Figure 6 sensors-19-04354-f006:**
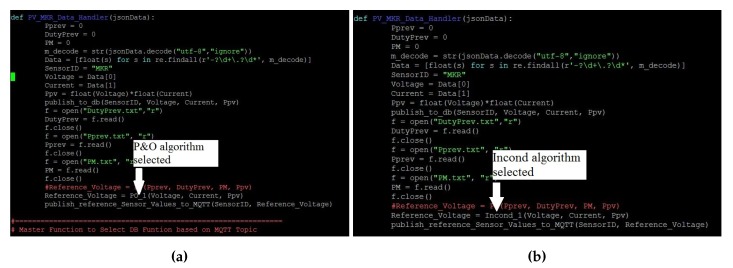
The MPPT algorithm selection. (**a**) P&O; (**b**) InCond.

**Figure 7 sensors-19-04354-f007:**
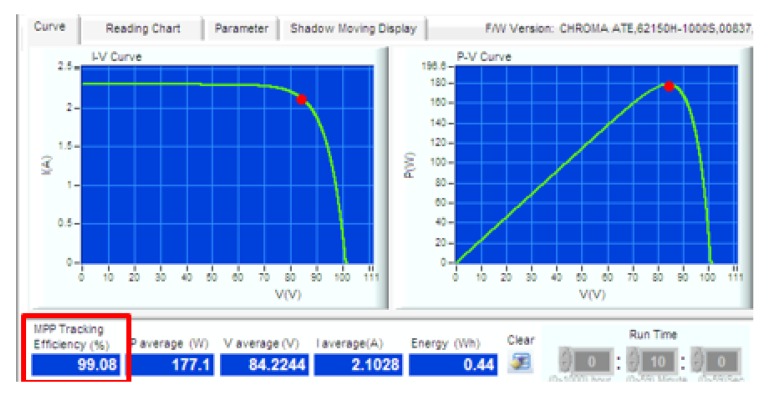
*I-V* curve and *P-V* curve of 334 W/m${}^{2}$ setting.

**Figure 8 sensors-19-04354-f008:**
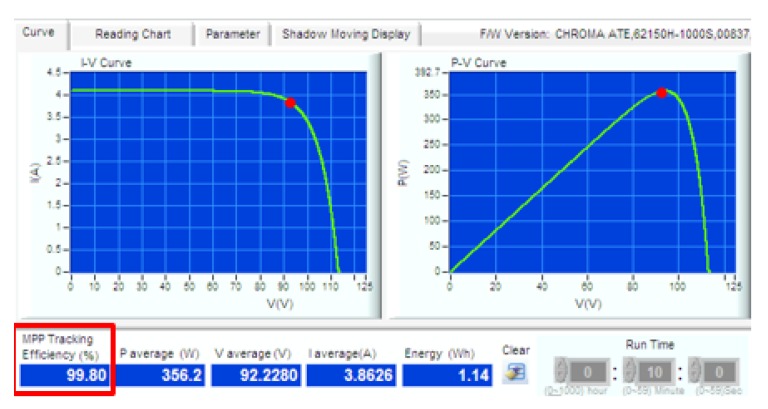
*I-V* curve and *P-V* curve of 667 W/m${}^{2}$ setting.

**Figure 9 sensors-19-04354-f009:**
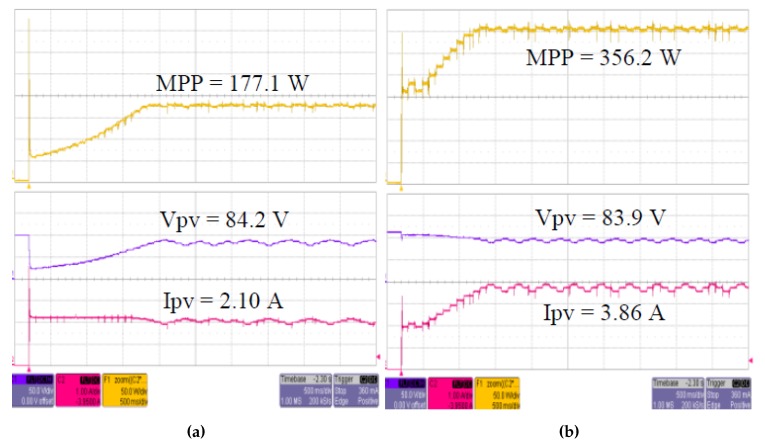
Tracking result of: (**a**) 334 W/m${}^{2}$ setting; and (**b**) 667 W/m${}^{2}$ setting.

**Figure 10 sensors-19-04354-f010:**
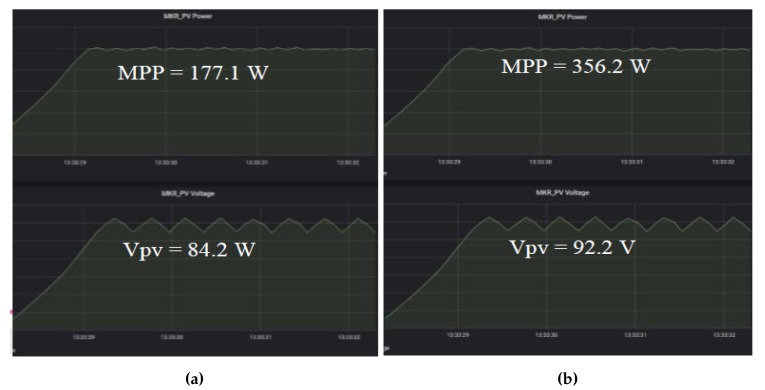
Data visualization on *Grafana*: (**a**) 334 W/m${}^{2}$ setting; and (**b**) 667 W/m${}^{2}$ setting.

**Figure 11 sensors-19-04354-f011:**
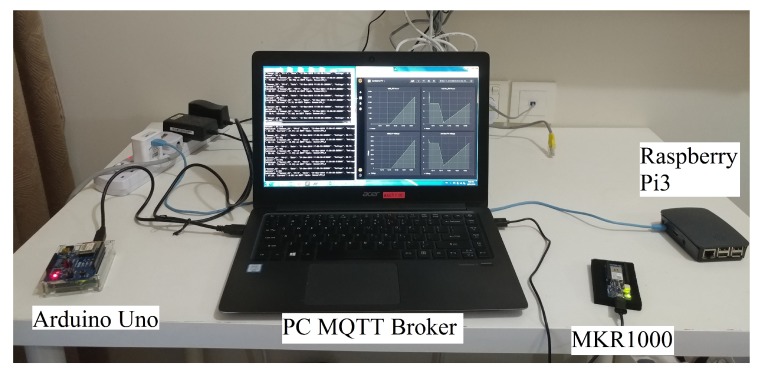
Experiment setup for system heterogeneity experiment.

**Figure 12 sensors-19-04354-f012:**
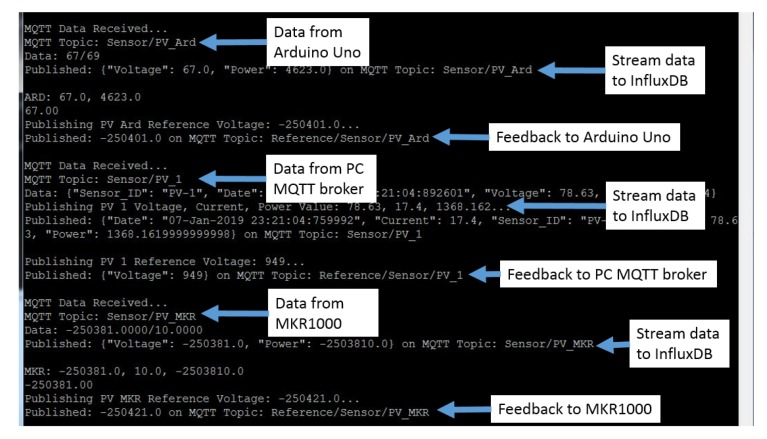
The data transmission between MKR1000, Arduino Uno and PC MQTT broker with Raspberry Pi3 and *InfluxDB*.

**Figure 13 sensors-19-04354-f013:**
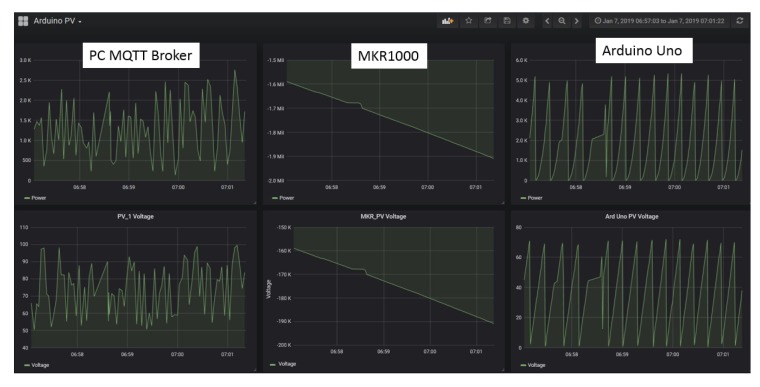
The data visualization of PC MQTT Broker, MKR1000 and Arduino Uno through *Grafana*.

**Figure 14 sensors-19-04354-f014:**
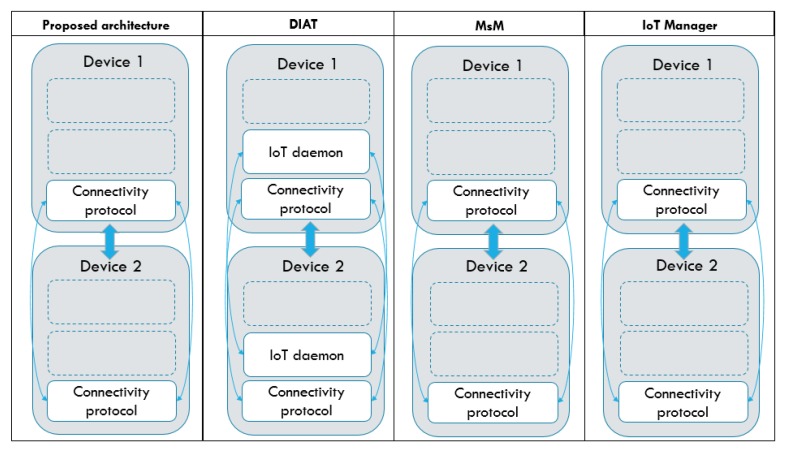
Minimum requirement for system interoperation for proposed architecture, DIAT and CEB Atlas.

**Figure 15 sensors-19-04354-f015:**
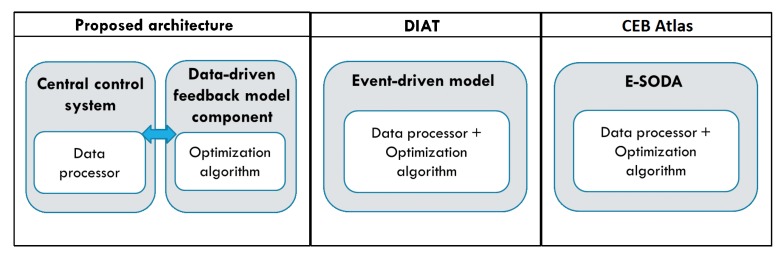
Data-driven function flexibility comparison between proposed architecture, DIAT and CEB Atlas.

**Table 1 sensors-19-04354-t001:** Comparisons of different IoT architectures.

	Data-driven Function	System	System
	Availability	Interoperability	Reusability
DIAT [15]	Yes	No	Yes
Condense [14]	No	No	Yes
Atlas CEB [16]	Yes	No	Yes
Predescu’s model [33]	Yes	No	No
Al-Ali’s model [34]	No	No	No
2EA [31]	Yes	No	No
FIFu [12]	No	No	No
MsM [32]	No	Yes	Yes
IoT-CANE [11]	No	No	No
IoT Manager [18]	No	Yes	Yes

**Table 2 sensors-19-04354-t002:** Parameters of KC85T solar PV module.

Solar PV Isolation	344 W/m${}^{2}$	667 W/m${}^{2}$
Voltage at MPP (Vmpp)	83.9 V	93.8 V
Current at MPP (Impp)	2.13 A	3.81 A
Open circuit Voltage (Voc)	101.2 V	113.2 V
Short circuit Current (Isc)	2.29 A	4.10 A

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
