# Peer review of "An Interoperable Component-Based Architecture for Data-Driven IoT System"

_sensors, 2019, doi:10.3390/s19204354_

Round 1

Reviewer 1 Report

Pros

Authors focus on the primal challenges of IoTs that affects  the services depending on the  operating systems and protocols. Authors major  contribution is the architecture that utilizes the component-based design approach to create and define the
loosely-coupled, standalone but interoperable service components for disparate IoT systems.  Three main challenges authors try to model is interoperability, system reusability
and the lack of data-driven functionality issues. Authors provide a proof-of-concept
prototype and evaluate the same.

Cons

The paper does not satisfy the requirements of MDPI journal as a whole due to its incomplete references. The reviewer highly recommends citing recent research trends such energy harvesting based techniques and the impact of IoT in 5G networks. These are really important considerations for future networks.

Balasubramanian, V., et al. "A unified architecture for integrating energy harvesting IoT devices with the Mobile Edge Cloud." 2018 IEEE 4th World Forum on Internet of Things (WF-IoT). IEEE, 2018,

V. Balasubramanian, F. Zaman, M. Aloqaily, I. A. Ridhawi, Y. Jararweh and H. B. Salameh, "A Mobility Management Architecture for Seamless Delivery of 5G-IoT Services," ICC 2019 - 2019 IEEE International Conference on Communications (ICC), Shanghai, China, 2019, pp. 1-7.

Author Response

Point 1:

The paper does not satisfy the requirements of MDPI journal as a whole due to its incomplete references. The reviewer highly recommends citing recent research trends such as energy harvesting based techniques and the impact of IoT in 5G networks. These are really important considerations for future networks.

Response 1:

The suggested publications are reviewed and we found out that they are significant works and related to the research direction of this manuscript. Therefore, the revised manuscript has included the suggested publications. Please refer to the attachment.

Reviewer 2 Report

The authors of this paper present the design of an Internet of Things
(IoT) platform based on a component-based approach for service components. The main goal of the proposed architecture is to provide interoperabilty as well as reusability. To that end, the authors evaluate their proposal with a use case prototype. The proposed system is (very) well described and presented. Moreover, the validation includes an actual IoT prototype using Raspberry and Arduino components as well as a microcontroller.

The paper is well written and flows well. One general observation on the contribution is its very particular application, which can be seen as an advantage as well as a limiting inconvenient. The results demonstrate that the algorithms proposed are capable of providing the same performance as offline MPPT systems. The proven interoperability and reusability of the proposed platform may be compromised under different environment conditions. Thus, it would be highly desirable to add a brief discussion of potential different use cases where the proposal is promising and others where it is not recommended at all.

Author Response

Point 1: 

The paper is well written and flows well. One general observation on the contribution is its very particular application, which can be seen as an advantage as well as a limiting inconvenient. The results demonstrate that the algorithms proposed are capable of providing the same performance as offline MPPT systems. The proven interoperability and reusability of the proposed platform may be compromised under different environment conditions. Thus, it would be highly desirable to add a brief discussion of potential different use cases where the proposal is promising and others where it is not recommended at all.

Response 1:

The proposal suggested a generic, cross-domain application IoT system for diverse applications. Therefore, the proposed component-based system is loosely coupled which only requires minimal standardization in terms of protocols. As a result, other potential uses cases where the proposal is promising and those that are not, are differentiated by the available resources and infrastructure of the domain. The discussions regarding this issue are included in section 5.6 (Potential application domains & use-cases, lines 567-590). Please refer to the attachment
